# Salivary osmolality measured by MX3 hydration testing system demonstrates high reliability but limited validity in elite athlete hydration assessment

Stefan Pettersson[1,2]*, Lykke Tamm[2], Stig Mattsson[2], Anton Kalén[2,3]

1 Center for Health and Performance, Department of Food and Nutrition, and Sport Science, University of Gothenburg, Gothenburg, Sweden, 2 Swedish Olympic Committee, Sofiatornet, Olympiastadion, Stockholm, Sweden, 3 Department of Computer Science, Electrical and Space Engineering, Luleå University of Technology, Luleå, Sweden

* stefan.pettersson@ped.gu.se

## Abstract

Accurate assessment of hydration is essential for both athletic performance and health, yet remains challenging outside controlled settings. This study evaluated the validity and reliability of salivary osmolality (SOSM), measured using a hand-held osmometer (MX3 Hydration Testing System), as a marker of morning hydration status in elite athletes. A total of 230 paired fasted-morning saliva and urine sample sets from 118 Olympic-level athletes (57% female; age $27 \pm 5$ years; BMI $24 \pm 2$) were included in the analyses. Urine specific gravity (USG) served as the reference. Validity was examined using ICC(3,1), ROC analysis, and Cohen's κ, and test–retest reliability was assessed from three repeated SOSM readings from the same saliva sample. SOSM showed no meaningful correspondence with USG (ICC = 0.00; 95% CI [−0.13, 0.13]) and no meaningful classification agreement (κ = −0.02; 95% CI [−0.15, 0.10]). Diagnostic accuracy was poor across all examined threshold comparisons, with AUCs of 0.40–0.49, indicating chance-level to worse-than-chance discrimination, sensitivity 40–69%, and specificity 23–65%. By contrast, within-sample repeated SOSM measurements were highly consistent (ICC = 0.93; 95% CI [0.91, 0.94]), with a standard error of measurement of 7.9 mOsm and a minimal detectable change (MDC$_{90}$) of 18.4 mOsm. These findings indicate that the MX3 device is repeatable when reading the same saliva sample, but SOSM lacks validity as a stand-alone marker of first-morning hydration status in elite athletes under free-living conditions.

## Introduction

Monitoring hydration status is important for both athletic performance and health, but valid assessment outside laboratory settings remains challenging [1]. Traditional

**Data availability statement:** All relevant data are within the paper and its Supporting information files.

**Funding:** The author(s) received no specific funding for this work.

**Competing interests:** The authors have declared that no competing interests exist.

biomarkers such as plasma osmolality and urine indices, including urine specific gravity (USG), are useful but have important limitations in field settings. Plasma osmolality is tightly homeostatically regulated, usually changes only when fluid deficits become more pronounced, and requires venous blood sampling with laboratory analysis, making it impractical for routine athlete monitoring [2–4]. Urine indices are non-invasive and widely used, but they reflect the renal response to recent fluid and solute intake over the preceding hours rather than an immediate snapshot of total-body water, and may therefore lag behind rapid changes in hydration status [4–6].

Salivary osmolality (SOSM) has emerged as a promising, non-invasive hydration marker, as dehydration reduces salivary flow and increases solute concentration. In controlled physiological studies, SOSM has shown strong correlations with plasma osmolality during progressive dehydration [7,8]. Oliver et al. [7] studied 48 h of fluid restriction with additional exercise and rehydration phases, whereas Walsh et al. [8] examined acute exercise-induced dehydration, However, salivary markers remain sensitive to sampling conditions; Ely et al. [9] showed that recent oral fluid exposure can significantly affect SOSM values, highlighting the importance of standardized sampling protocols. This also complicates the direct translation of findings from pediatric and elderly clinical cohorts [10,11] to elite athletes, whose training and competition environments may involve factors such as oral breathing, carbohydrate intake, and repeated mouth rinsing that could influence saliva composition independently of whole-body hydration [9,12].

Standardized conditions—such as overnight fasting—minimize external influences like recent fluid intake, diet, and physical activity, making first-morning urine a widely accepted reference for hydration assessment [13]. In athlete monitoring, this timing is particularly relevant because it reflects hydration status at the start of the day, before substantial behavioral and environmental variation accumulates. It is routinely used in both research and clinical settings, and leading sports organizations recommend first-morning urine or body mass monitoring to prevent athletes from beginning training in a hypohydrated state [13–16].

For field-based applications, a portable salivary osmometer (MX3 Hydration Testing System (HTS), MX3 Diagnostics Inc., Melbourne, Australia) has recently been developed, enabling spot hydration checks via immediate SOSM readings. Studies using the MX3 HTS in non-athletic populations have shown promising results. For example, Atjo et al. [10] reported a strong correlation (r = 0.78) between SOSM and USG in older adults with hypertension, and Faidah et al. [11] found similar agreement in a pediatric population. However, these populations differ substantially from elite athletes in age, health status, and medication use, which limits direct generalization to highly trained athlete populations. The validity and reliability of the MX3 HTS for first-morning hydration assessment in elite athletes therefore require direct evaluation.

The aim of this study was to evaluate the validity and reliability of SOSM, measured using the MX3 HTS, as a marker of first-morning hydration status in elite athletes. Specifically, we aimed to (1) examine the agreement between SOSM and first-morning USG under free-living conditions and (2) assess the within-sample

test–retest reliability of SOSM under overnight fasted conditions. The findings may help define the practical utility of SOSM for athlete hydration monitoring.

## Participants and recruitment

The participating athletes in this study were part of the Swedish Olympic Committee's Top and Talent support program and represented the national team level. As shown in Table 1, a total of 118 athletes (51 males and 67 females; age 27±5 years) from 14 different sports participated (a full list of sports is provided in the Supplementary Table S1 in S1 File). To classify athletes' training and competition level, participants were categorized using the framework by McKay et al., [17] which defines five competitive tiers. In our sample, 34% (n=40), including 26 team sport athletes, were classified as World-Class (Tier 5), representing finalists or top-ranked athletes at Olympic Games or World Championships, with near-maximal training volumes and exceptional sport-specific skills. Another 64% (n=75) were categorized as Elite (Tier 4), competing internationally or in top professional leagues, training at maximal or near-maximal levels. The remaining 2% (n=3) were classified as Highly Trained (Tier 3), indicating national-level athletes performing structured, periodized training. This distribution across sport categories and competitive tiers is summarized in Supplementary Table S2 in S1 File. Recruitment and data collection were conducted prospectively from 5 February 2024–7 November 2024. The majority of data were collected during pre-Olympic training camps held in Fuerteventura, Spain, in February 2024 (various sports) and June 2024 (handball), in Font-Romeu, France, in October 2024 (biathlon), and in Stockholm, Sweden, in November 2024 (ice hockey). Prior to enrollment, all participants provided written informed consent. The study was approved by the Swedish Ethical Review Authority (ref number 2023-06727-01).

## Procedures

### Overall study design

In accordance with the American College of Sports Medicine (ACSM) guidelines for hydration assessment [13], each participant was instructed to provide a midstream urine sample from their first morning void between 06:00 and 08:00 AM. To facilitate this, participants were provided with a sterile urine specimen container the evening prior to sampling. No

**Table 1. Descriptive and sport characteristics of the 118 participants (mean±SD). Relative values (%) are shown in parentheses.**

|  | Female (n=67) | Male (n=51) | All athletes (n=118) |
|---|---|---|---|
| Age (years) | 25±4 | 28±5* | 27±5 |
| Height (m) | 1.73±0.07 | 1.85±0.08* | 1.78±0.09 |
| Weight (kg) | 69±7 | 85±13* | 76±12 |
| BMI (kg) | 23±2 | 25±2* | 24±2 |
| Sport classification |  |  |  |
| Endurance | 39 (58%) | 17 (33%) | 56 (47%) |
| Middle distance/Power | 8 (12%) | 6 (12%) | 14 (12%) |
| Precision/Skill | 6 (9%) | 7 (14%) | 13 (11%) |
| Speed/Strength | 3 (4%) | 4 (8%) | 7 (6%) |
| Combat Sports | 2 (3%) | 4 (8%) | 6 (5%) |
| Sport level |  |  |  |
| Tier 3 | 0 (0%) | 3 (6%) | 3 (2.5%) |
| Tier 4 | 45 (67%) | 30 (59%) | 75 (64%) |
| Tier 5 | 22 (33%) | 18 (35%) | 40 (34%) |

*Note*: BMI; Body Mass Index. *Significant (p<0.001) differences between female and male athletes.

standardized diet or fluid-intake protocol was imposed on the preceding day; participants were asked to follow their normal routines, and no attempt was made to manipulate hydration status prior to sample collection so that the measurements would reflect free-living morning conditions.

To ensure standardization, participants were instructed to refrain from eating or drinking, brushing teeth, rinsing the mouth, or using moist oral tobacco (e.g., snus, placed under the upper lip) from the time they went to bed the night before until saliva collection was completed. No formal information was collected on nocturnal mouth breathing, snoring, nasal congestion, xerostomia, or other potential causes of overnight oral drying. Within 15 minutes of (morning) urine collection, participants handed the sample container to a member of the research team. At that same time, an unstimulated saliva sample (3–5 mL) was collected by expectoration into a sterile sampling tray, whereby participants were instructed to allow saliva to accumulate in the mouth before gently expelling it. Adherence to fasting instructions between urine and saliva collection was self-reported and not independently verified.

## Sample analysis

### Urine

Urine samples were analyzed for USG using a digital refractometer with automatic temperature compensation (Atago PAL-10S, Tokyo, Japan; range: 1.000–1.060; resolution: 0.001). Refractometry provides an indirect estimate of USG by measuring the refractive index, which reflects solute concentration [18]. A 0.3 mL aliquot was pipetted onto the refractometer prism three times in succession, with recalibration (zeroing with distilled water) performed once per participant.

### Saliva

SOSM was measured in milliosmoles (mOsm) using the MX3 HTS (MX3 Diagnostics Inc., Melbourne, Australia). The method is based on electrochemical impedance spectroscopy and utilizes pre-calibrated disposable biosensor strips. The same saliva sample was used for three consecutive measurements with new biosensor strips for each reading. All analyses (urine and saliva) were performed within 30 minutes of sample collection, and specimens were discarded immediately thereafter.

### Hydration classification

The study did not aim to diagnose hypohydration per se, but rather to evaluate agreement between SOSM values and USG at three levels: continuous values, binary dehydration thresholds, and independently assigned hydration classifications based on the MX3 HTS manufacturer's guidance [19] and the USG thresholds proposed by Casa et al. [20]. Given the lack of scientific consensus regarding the most appropriate USG cut-off for hydration assessment, thresholds of 1.020, 1.025, and 1.030 were examined, as these have all been used in the hydration literature and practice [2,4,14,20]. Corresponding SOSM thresholds were determined empirically from the data. Because USG and SOSM reflect different physiological processes, the comparison was intended to assess empirical alignment between field-based classification systems rather than biological equivalence between markers.

Hydration status was categorized into four levels according to the MX3 HTS manufacturer's guidance for SOSM [19] and the USG thresholds proposed by Casa et al. [20]. For SOSM, categories were hydrated (<66 mOsm), mild dehydration (66−100 mOsm), moderate dehydration (101−150 mOsm), and severe dehydration (≥151 mOsm). The corresponding USG categories were well hydrated (≤1.010), minimal dehydration (1.011–1.020), significant dehydration (1.021–1.030), and serious dehydration (≥1.031).

### Power calculation

To ensure adequate precision for the planned ICC-based validity analysis, a minimum of 128 paired SOSM/USG measurements was required [21]. The power analysis was conducted using the ICC.Sample.Size package (version 1.0) in R

(version 4.3.0), following equations outlined by Zou [22]. The target sample size referred to paired measurement occasions rather than unique athletes, and the final analytic sample exceeded this target.

## Statistics

Descriptive statistics are presented as mean ± standard deviation (SD) for continuous variables and frequencies (%) for categorical variables. To assess the validity of SOSM against USG, the intraclass correlation coefficient (ICC(3,1)) was calculated with 95% confidence intervals (CIs). Because some athletes contributed repeated morning sample sets, analyses were conducted at the sample-set level. No mixed-effects model or other adjustment for within-subject clustering was applied. ICC values were interpreted according to Koo and Li [23] as poor (<0.50), moderate (0.50–0.75), good (0.75–0.90), and excellent (>0.90).

Receiver operating characteristic (ROC) curves were used to identify optimal SOSM thresholds corresponding to USG thresholds (1.020, 1.025, and 1.030), optimizing Youden's index. Area under curve (AUC), sensitivity, and specificity for resulting SOSM thresholds were reported. Cohen's kappa (κ) was used to evaluate agreement between USG- and SOSM-based classifications, with interpretation according to McHugh [24] as none (<0.20), minimal (0.21–0.39), weak (0.40–0.59), moderate (0.60–0.79), strong (0.80–0.90), and almost perfect (>0.90). Weighted kappa was applied for ordered hydration categories. Bootstrap resampling with 10,000 iterations was utilized to estimate 95% CIs. Given the repeated-measures structure, interpretation focused primarily on the magnitude and direction of the agreement and accuracy metrics (e.g., ICC near 0, κ near 0, AUC near or below 0.50) rather than on statistical significance alone.

Test-retest reliability of SOSM was assessed using same-day repeated measures and ICC(3,1). In addition, the Standard Error of Measurement (SEM) and Minimal Detectable Change at 80% ($MDC_{80}$) and 90% ($MDC_{90}$) CI were calculated to assess measurement precision. Bland-Altman plots and within-day intra-individual coefficients of variation (CVs) were also used to further evaluate test-retest reliability. All statistical analyses were conducted using R (version 4.5.0; released 2025-04-11). Statistical significance was defined as $p < 0.05$.

## Results

A total of 230 paired first-morning saliva and urine sample sets from 118 athletes were included in the final analysis; 13 additional sampling events were excluded because the MX3 device did not return a readable SOSM value. Of the 118 athletes, 49 (42%) contributed sample sets from one morning, 26 (22%) from two mornings, and 43 (36%) from three mornings. Across the 230 paired sample sets, mean first-morning USG was 1.019 ± 0.005 (range 1.005 to 1.030), and mean SOSM was 93 ± 29 mOsm (range ~30–120 mOsm). Descriptive characteristics of the sample sets are presented in Table 2. No significant (p > 0.05) sex difference in either USG or SOSM were observed (Table 2).

### Validity of SOSM against USG

The relationship between SOSM and USG is shown in Fig 1. SOSM showed poor consistency with USG (ICC(3,1) = 0.00, 95% CI [−0.13, 0.13]).

### SOSM threshold

Table 3 summarizes the diagnostic performance of SOSM thresholds against the corresponding USG thresholds. Across all examined thresholds, diagnostic accuracy was poor, with AUC values ranging from 0.40 to 0.49, indicating discrimination no better than or worse than chance. Sensitivity and specificity were also low across thresholds, and Cohen's kappa values remained close to zero, indicating no meaningful agreement between SOSM- and USG-based classifications.

**Table 2. Descriptive characteristics of fasted morning sample sets (N = 230) collected from 118 athletes. Mean values (± SD) and dehydration classifications are shown; proportions within each subgroup (%) are shown in parentheses.**

| | Female (n = 67) | Male (n = 51) | All athletes (n = 118) |
|---|---|---|---|
| Total number of USG±SOSM sets | 98 | 132 | 230 |
| 1 day | 44 (66) | 5 (10) | 49 (42) |
| 2 days | 15 (22) | 11 (21) | 26 (22) |
| 3 days | 8 (12) | 35 (69) | 43 (36) |
| USG | 1.019 ± 0.006 | 1.020 ± 0.005 | 1.019 ± 0.005 |
| USG condition | | | |
| Well hydrated | 3 (3) | 1 (1) | 4 (2) |
| Minimal dehydration | 54 (55) | 72 (55) | 126 (55) |
| Significant dehydration | 39 (40) | 56 (42) | 95 (41) |
| Serious dehydration | 2 (2.0) | 3 (2) | 5 (2.2) |
| USG threshold | | | |
| Dehydrated (USG > 1.020) | 41 (42) | 59 (45) | 100 (43) |
| Dehydrated (USG > 1.025) | 10 (10) | 19 (14) | 29 (13) |
| Dehydrated (USG > 1.030) | 2 (2.0) | 3 (2) | 5 (2) |
| SOSM (mOsm) | 89 ± 27 | 97 ± 30 | 93 ± 29 |
| SOSM condition | | | |
| Hydrated | 15 (16) | 13 (11) | 28 (13) |
| Mildly dehydrated | 56 (60) | 69 (56) | 125 (58) |
| Moderately dehydrated | 21 (22) | 36 (29) | 57 (26) |
| Severely dehydrated | 2 (2.1) | 5 (4) | 7 (3.2) |

*Note:* USG; urine specific gravity, SOSM; salivary osmolality. Values for USG > 1.020, > 1.025, and >1.030 represent cumulative binary thresholds and are therefore not mutually exclusive.

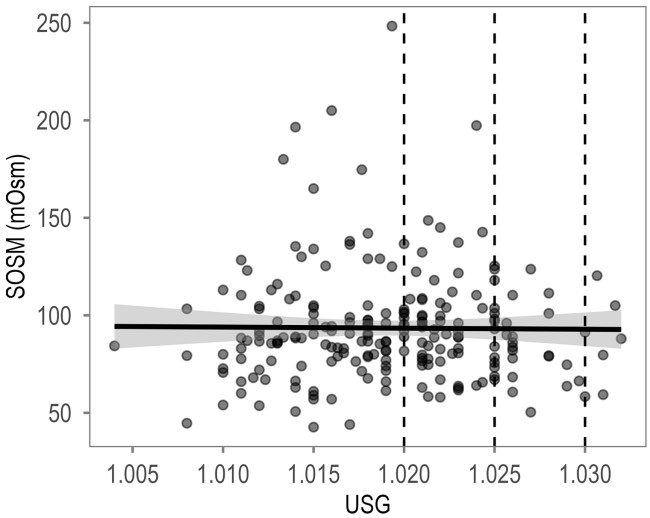

**Fig 1. Relationship between SOSM and USG.** Correlation (black line) with 95% confidence interval (gray area). Vertical lines indicate different USG thresholds for dehydration as suggested by Casa et al. [20].

**Table 3. Diagnostic performance and agreement between USG and SOSM thresholds in detecting hydration status (N = 230 paired sample sets).**

| USG threshold | AUC | SOSM threshold | Sensitivity (%) | Specificity (%) | Cohens kappa | |
|---|---|---|---|---|---|---|
| | | | | | κ | 95% CI |
| 1.020 | 0.49 | 84 | 57 | 37 | −0.06 | [−0.18, 0.07] |
| 1.025 | 0.40 | 76 | 69 | 23 | −0.04 | [−0.10, 0.03] |
| 1.030 | 0.49 | 97 | 40 | 65 | 0.01 | [−0.05, 0.06] |

*Note:* USG; urine specific gravity, SOSM; salivary osmolality.

## Cross-classification

Cross-classification further demonstrated substantial discordance between methods. Among the 89 samples classified as significantly dehydrated by USG, 12 (13%) were classified as hydrated by SOSM, while most were classified as only mildly dehydrated. Overall agreement between ordinal USG and SOSM categories was not meaningful (κ = −0.02, 95% CI [−0.15, 0.10]). Detailed cross-classification data are provided in Supplementary Table S3 in S1 File.

## Test-retest SOSM reliability

As illustrated in Fig 2, the test–retest reliability of intra-day SOSM measurements was excellent (ICC(3,1) = 0.93, 95% CI [0.91, 0.94]). These repeated within-sample measurements reflect device repeatability, that is, measurement consistency when the same saliva sample is read repeatedly under standardized conditions. The standard error of measurement was 7.9 mOsm, and the minimal detectable change was 14.3 mOsm at the 80% confidence level ($MDC_{80}$) and 18.4 mOsm at the 90% level ($MDC_{90}$), meaning that repeated readings must differ by at least this amount to exceed measurement error. The Bland–Altman plot showed a mean difference close to zero, with no apparent proportional bias across the measurement range.

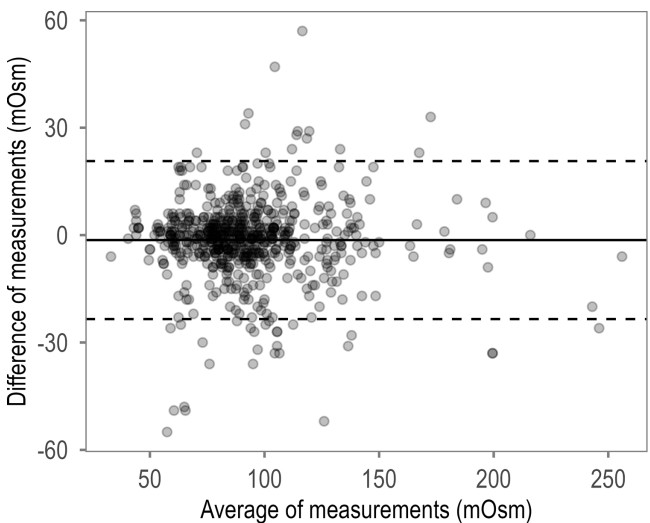

**Fig 2. Bland–Altman plot illustrating the agreement between same-day repeated measurements of salivary osmolality (SOSM).**

## Discussion

While the performance of the MX3 HTS for measuring SOSM has previously been evaluated against USG in healthy pediatric [11] and elderly clinical populations [10], no study to date has, to the authors' knowledge, investigated its application in elite athletes with a concurrent assessment of both criterion validity and test–retest reliability.

Our findings demonstrated that SOSM showed poor validity as a marker of morning hydration status when compared to USG. No meaningful relationship was observed between SOSM and USG (r = −0.01, 95% CI [−0.14, 0.12]; ICC = 0.00, 95% CI [−0.13, 0.13]), and classification agreement across hydration categories was not meaningful (Cohen's κ = −0.02; 95% CI [−0.15, 0.10]). In contrast, Atjo et al. [10] reported a strong correlation (r = 0.78, p < 0.001) between SOSM and first morning USG in a clinical cohort, achieving high sensitivity and specificity at specific SOSM thresholds. In our sample, however, classification performance was markedly lower across all thresholds. For USG ≥ 1.025, a SOSM cut-off of 76 mOsm yielded 69% sensitivity but only 23% specificity. At USG ≥ 1.030, sensitivity dropped further to 40% and specificity to 65%, with κ-values remaining close to zero. AUC values below 0.50 indicate that SOSM did not discriminate higher versus lower USG states better than chance in this sample. One contributing factor may be the lower prevalence of dehydration in our athletic cohort—13% exceeded the USG threshold of 1.025 and 2% exceeded 1.030—compared with 89% and 70% in the clinical sample reported by Atjo et al. [10] in which 47% (n = 70) were diuretic users. Previous research indicates that markers like USG perform better under conditions of more pronounced dehydration, where physiological signals are less masked by biological variability [3,5,6]. Similar limitations likely apply to SOSM, as its diagnostic utility diminishes at lower fluid deficits due to reduced signal-to-noise ratio [8,12]. This highlights the challenge of validating hydration biomarkers in cohorts such as the present one, where hydration status is generally characterized by relatively small deviations from euhydration.

Several physiological and methodological factors may help explain the weak correspondence between SOSM and USG. Circadian changes in salivary composition, such as decreased saliva flow and elevated osmolality upon waking, may artificially inflate morning SOSM values [25,26]. This interpretation is supported by our classification data: although only 43% of collected samples sets exceeded the USG dehydration threshold (>1.020), 87% of SOSM readings were categorized as mildly to severely dehydrated. Overnight xerostomia, mouth breathing, snoring, or nasal obstruction could further concentrate oral fluid independently of whole-body hydration, and these factors were not assessed in the present study. Practically, this means that some athletes with elevated first-morning USG could be classified as hydrated by SOSM, potentially providing false reassurance in field settings. Cross-tabulations further revealed substantial misclassification between the two methods. Additionally, 13 data sets were excluded from analysis because the MX3 device did not return a readable SOSM value. If highly viscous or concentrated saliva was disproportionately unreadable, this could introduce selection bias by removing exactly those cases in which the device might be most challenged under field conditions. Elevated saliva viscosity is a known limitation that can impair measurement accuracy by reducing effective sensor contact with the sampling matrix [19]. Taken together, these observations suggest that morning SOSM may reflect transient oral or circadian concentration effects rather than true whole-body hydration status, particularly under fasted, free-living conditions.

In contrast to the findings of limited validity, SOSM measured using the MX3 HTS displayed excellent test–retest reliability under standardized measurement conditions. The average intra-individual coefficient of variation (CV) across repeated measurements from the same saliva sample was 6.4 ± 6.7%, ranging from 0.0% to 59.9%, indicating modest biological and/or procedural variability. Atjo et al. [10] reported similar mean intra-sample CV (7 ± 6%) for the MX3 HTS, although they did not evaluate test–retest reliability directly. Winter et al. [27] reported moderate within-day reliability, with an ICC of 0.75 and an MDC$_{90}$ of 21.3 mOsm, based on two separately collected saliva samples. Our higher ICC of 0.93 and lower MDC of 14.4 mOsm and 18.5 mOsm at the 80% and 90% confidence levels, respectively, likely reflect improved precision when repeated measurements are taken from the same sample, thereby minimizing variation from collection technique or transient oral factors. Thus, while Atjo and colleagues and Winter et al. provide relevant insight into real-world field application, our data demonstrate the upper limit device repeatability under standardized conditions.

USG was selected because first-morning urine is a widely used practical field-based hydration index, whereas plasma osmolality was not feasible in the present camp-based setting. Although USG served as the reference method, it is not without limitations, particularly in athletic populations. USG can be influenced by recent protein intake, muscle mass, and training stress [28]. Moreover, the magnitude and even presence of associations with body composition appear to depend on sampling-time: such associations are absent in fasted first-morning samples but present in non-fasted spot samples later in the day [29]. Population-based analyses have reported higher odds of elevated USG among individuals with greater BMI and lean body mass [30–32]. In our cohort, BMI values were within the normal range (24±2), and first morning USG (1.019±0.005) aligned with population-based data; together with prior studies supporting first-morning USG as a valid field index, these observations make body-size confounding an unlikely explanation for the poor agreement between SOSM and USG. Nonetheless, the present design cannot determine whether the observed discordance reflects failure of SOSM, limitations of USG, or some degree of error in both methods.

Several study limitations should be acknowledged. First, diet and fluid intake on the preceding day were not standardized, which preserved ecological validity but likely increased biological variability in both USG and SOSM. Furthermore, overnight fasting duration was not quantified, and adherence to fasting instructions was not formally verified beyond participant instruction. Second, participants were not screened for mouth breathing, snoring, nasal obstruction, or subjective dry mouth, all of which may influence morning saliva concentration independently of hydration status. Third, some athletes contributed repeated morning observations, meaning that the data were not fully independent, and no mixed-effects model was applied. This should be considered when interpreting the precision of the estimates. This may have led to somewhat optimistic precision estimates, but is unlikely to explain the overall pattern of near-zero agreement and poor discrimination. Fourth, neither plasma osmolality nor body mass change was assessed, which precludes definitive conclusions about whether the observed discordance reflects limitations of SOSM, USG, or both. Finally, unreadable SOSM results were not random by definition and may represent a practical failure mode of the device under real-world field conditions.

## Conclusion and practical implications

Our findings demonstrate that SOSM, measured using the MX3 HTS, shows poor validity as an indicator of morning hydration status in elite athletes. SOSM exhibited no meaningful relationship with USG, low diagnostic accuracy and no meaningful classification agreement. These results suggest that under fasted, morning conditions, SOSM may reflect transient oral or circadian effects rather than whole-body hydration status. In contrast, SOSM showed excellent test–retest reliability across repeated measurements from the same sample, indicating high measurement stability under standardized procedures. Although USG served as the reference in this study, its accuracy in reflecting true hydration status, particularly in athletic populations, was not directly evaluated. Future studies should examine SOSM under other sampling conditions before any practical role can be inferred.

## Supporting information

**S1 File. Supplementary materials: This file contains three supplementary tables: Table S1, categorization of included sports by sport type and number of participants; Table S2, distribution of participants across sport categories and performance tiers; and Table S3, cross-classification of hydration status based on USG and SOSM categories. S1 Dataset (sosm_usg_dataset.csv).**
(DOCX)

## Acknowledgments

The authors wish to thank the athletes who volunteered to participate in this study.

## Author contributions

**Conceptualization:** Stefan Pettersson, Lykke Tamm, Stig Mattsson, Anton Kalén.

**Data curation:** Stefan Pettersson, Stig Mattsson, Anton Kalén.

**Formal analysis:** Stefan Pettersson, Anton Kalén.

**Investigation:** Stefan Pettersson, Stig Mattsson, Anton Kalén.

**Methodology:** Stefan Pettersson, Lykke Tamm, Stig Mattsson, Anton Kalén.

**Project administration:** Stefan Pettersson.

**Resources:** Stefan Pettersson.

**Supervision:** Stefan Pettersson.

**Visualization:** Stefan Pettersson, Anton Kalén.

**Writing – original draft:** Stefan Pettersson.

**Writing – review & editing:** Stefan Pettersson, Lykke Tamm, Stig Mattsson, Anton Kalén.

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
