## [Decision Letter · Decision Letter 0]

6 Apr 2026

PONE-D-25-59162Salivary Osmolality Measured by MX3 Hydration Testing System Demonstrates High Reliability but Limited Validity in Elite Athlete Hydration AssessmentPLOS One

Dear Dr. Pettersson,

Thank you for submitting your manuscript to PLOS ONE. After careful consideration, we feel that it has merit but does not fully meet PLOS ONE’s publication criteria as it currently stands. Therefore, we invite you to submit a revised version of the manuscript that addresses the points raised during the review process.

We look forward to receiving your revised manuscript.

Kind regards,

Ozkan Isik

Academic Editor

PLOS One

Journal Requirements:

Reviewers' comments:

Reviewer's Responses to Questions

**Comments to the Author**

1. Is the manuscript technically sound, and do the data support the conclusions?

Reviewer #1: Yes

Reviewer #2: Yes

Reviewer #3: Partly

Reviewer #4: Yes

Reviewer #5: Yes

2. Has the statistical analysis been performed appropriately and rigorously? 

Reviewer #1: Yes

Reviewer #2: Yes

Reviewer #3: Yes

Reviewer #4: Yes

Reviewer #5: Yes

3. Have the authors made all data underlying the findings in their manuscript fully available?

Reviewer #1: Yes

Reviewer #2: Yes

Reviewer #3: Yes

Reviewer #4: Yes

Reviewer #5: Yes

4. Is the manuscript presented in an intelligible fashion and written in standard English?

Reviewer #1: Yes

Reviewer #2: Yes

Reviewer #3: Yes

Reviewer #4: Yes

Reviewer #5: Yes

5. Review Comments to the Author

Reviewer #1: To Whom It May Concern:

Thank you for this work comparing hydration assessment tools. This work is very informative and important for athletes and those working with athletes and workers in hot conditions. There are some concerns presented below. Please address these concerns, and this manuscript may be fit for publication.

Comments

I think an important consideration is that this study assessed two different measurement tools to assess two different types of body tissue/fluid. Therefore, you are simultaneously testing the accuracy of the devices as well as the usefulness of one body fluid to reflect the other (or the other’s hydration category). For instance, there was high ICC for test-retest within the same sample, indicating reliability of the device. I think another important consideration is the reliability of saliva, itself, to categorize hydration status. Based on these results, it might be more appropriate to label saliva as inconsistent for showing hydration status. In other words, if the device was used to test urine, would it have closer agreement to the USG categorization?

Abstract

L30-31: A kappa value <0 might be better described as no agreement rather than negligible.

L74: You mention that an aim to this study is to assess the test-retest reliability of SOSM under overnight fasted conditions. Did you confirm participants were fasted or did you control the amount of time fasted? If you did confirm this or have an average amount of time fasted, this information can go around L123-126.

Introduction

L150: Period appears before the reference.

L160: Period appears before the reference.

Methods

Hydration Classification

This needs to be clarified better because there is some confusion. First, you list the categories hydrated, mild dehydration, moderate dehydration, and severe dehydration as having both SOSM and USG criteria. Does a participant need to fit both criteria for the classification?

Based on Table 2, these appear to only be the titles of the SOSM classification. Please define the categories for USG: well hydrated, minimal dehydration, significant dehydration, and serious dehydration.

Results

Table 2: In the male column, the sub-group proportions add up to 100.8% instead of 100%.

The same problem is true for some of the sub-groups for USG condition and for SOSM condition.

It is also confusing that there are two categorizations for USG condition. This makes sense that the USG and SOSM may have different categories, but it’s not clear why USG has both well hydrated to serious dehydration as well as the dehydrated categories (>1.020, >1.025, and >1.030). For dehydration categories, were individuals in the >1.030 category also counted for the >1.025 and >1.020 categories? If not, then it is suggested that the categories are better defined (e.g. 1.025 ≥ USG > 1.020 and 1.030 ≥ USG > 1.025). Of these classified >1.020, are these the same individuals that are categorized in the moderate and severe dehydration groups? Please provide more clarity here.

L229-230: Please define “minimal” and “significant dehydration” in the methods section.

Discussion

With the SOSM threshold for 1.025 described as 76 mOsm, falling below the 84 mOsm for 1.020 and 97 mOsm for 1.030, can you speculate or clarify why this would be lower?

L258: I wonder if you can make equal comparisons between studies. For instance, what was the r value between USG and SOSM for the present study?

L265: You may want to re-phrase this as USG categories rather than cut-offs since this suggests a threshold, and therefore both groups would count within the 1.025 USG cut-off (i.e. minimum value as opposed to a range 1.025-1.030).

Since you found specific MDC80 and MDC90 thresholds, can you explain what this means to the average user of these devices?

L305: Change “was” to “were”.

Limitation

Why was serum not analyzed. Comparing an estimate (with inherent errors) to another estimate (with inherent errors) is very difficult to interpret what is accurate. Was this unattainable?

Reviewer #2: A well written manuscript of a well performed study. the sample of participants is high: however it is unclear whether there have been any missing values or participants willing to join who actually did drop out.

The differences in weather conditions in Spain (Feb), France (June) and Sweden (Oct) have not been taken into account. Could any changes in the environmental situation influence the (adaptation to) the saliva osmolarity differently than the urine sg? Are the results for samples taken at the three locations identical?

IN the Result section you write (lines 190-1) that there are no sex difference in USG. This seems also the case for the SOSM, so why not add it.

Reviewer #3: Dear Author

The article evaluates the usefulness of salivary osmolality (SOSM), measured with the portable MX3 system, as an indicator of hydration status in elite athletes. The results demonstrated very high test–retest reliability but no meaningful correlation with urine specific gravity (USG), indicating limited diagnostic validity of this method under real-world morning conditions. The authors conclude that SOSM should not be used as a stand-alone tool for hydration assessment despite its high measurement stability.

1. Why was urine specific gravity selected as the sole reference method, given that it is not considered the gold standard for hydration assessment?

2. Did the authors consider including plasma osmolality or body mass changes as additional validation markers?

3. How was the potential influence of circadian rhythm on salivary osmolality controlled, especially since measurements were performed exclusively in the morning?

4. Could the lack of intentional hydration manipulation among participants have limited the range of observed values and thereby affected the diagnostic validity of SOSM?

5. How do the authors interpret the finding that most samples were classified as dehydrated according to SOSM, while USG indicated normal or only mildly reduced hydration status?

6. Is it possible that discrepancies between SOSM and USG arise from physiological differences in saliva and urine regulation rather than purely methodological limitations?

7. Did the authors conduct subgroup analyses (e.g., by sex, sport discipline, or performance level) to determine whether SOSM validity differs across groups?

8. To what extent might saliva viscosity and unsuccessful device readings have influenced the final results, and was a sensitivity analysis performed after excluding these cases?

9. Are the SOSM classification thresholds provided by the device manufacturer appropriate for elite athletes, or should they be calibrated specifically for this population?

10. How do the authors envision potential applications of SOSM under alternative measurement conditions, such as immediately post-exercise or during training sessions?

11. Could repeated measurements from the same saliva sample have artificially inflated the reliability estimates compared to real-world use conditions?

12. What are the practical implications of these findings for coaches and medical staff who already use salivary osmolality devices in athlete monitoring?

Kind regards

Reviewer

Reviewer #4: 1. Overall Recommendation

• Decision: Major Revisions

• Originality and Contribution: Yes

• Scientific Significance: Yes

• General Comment: This study addresses a practical and relevant method for managing dehydration in elite athletes. While the study is well-structured and utilizes a high-level participant group (Olympic athletes), significant improvements are required in the methodology, statistical interpretation, and discussion of validity before publication

2. Detailed Evaluation and Revision Requirements

Abstract and Title

• Methodology: Briefly include the specific statistical analyses used within the abstract.

• Findings: The values of ICC = 0.00 and κ = -0.02 are critical. They suggest zero correlation between SOSM and the gold standard (USG). This needs to be highlighted clearly.

• AUC Values: The AUC range of 0.40–0.49 indicates predictive performance worse than chance. The abstract must reconcile this with the "excellent" test-retest reliability mentioned.

• Conclusion: The abstract focuses on reliability but lacks sufficient emphasis on the concerning lack of validity.

Introduction

• Methodological Limitations: Provide more detail on the underlying physiological causes behind the limitations of plasma osmolality and urine indices in field settings.

• Contextualization: Clarify if previous "controlled studies" were randomized trials or focused on acute/chronic phases. Specifically, explain why SOSM findings in pediatric/elderly populations (11, 12) do not translate to elite athletes (e.g., exercise-induced changes in saliva).

• Timing: Explicitly state the physiological rationale for choosing morning hydration measurements.

Methodology

• Dietary Control: Clarify if a dietary or fluid intake protocol was followed the previous day. If not, this must be listed as a limitation, as meal patterns impact morning hydration.

• Mouth Breathing: Address whether participants were screened for mouth breathing during sleep, which increases saliva viscosity and artificially inflates osmolality.

• Sample Power: The power calculation required 128 matched samples, but only 118 participants were included. Explain this shortfall or specify if any data were excluded.

• Statistical Independence: Since 230 samples were taken from 118 athletes (repeated measures), authors must confirm if a mixed-effects model was used to adjust for "within-subject" correlations.

• Threshold Comparison: Explain the biological overlap between the manufacturer's SOSM thresholds and the Casa et al. (15) USG references to justify the "negligible agreement" finding.

Findings

• Nested Data: Clarify in the statistics section how the data from participants providing multiple samples (36% provided 3 days of data) were handled to avoid violating the assumption of independent observations.

• Accuracy: Explicitly discuss why the AUC values (0.40–0.49) suggest the device is statistically less accurate than a random guess (0.50).

• Categorical Discordance: Address Table 4, where athletes labeled "significantly dehydrated" by urine analysis were classified as "hydrated" by the SOSM device.

Discussion

• Selection Bias: The exclusion of 13 data sets due to "high viscosity" is critical. If the most dehydrated athletes have the most viscous saliva and the device cannot read them, this creates a bias. This must be discussed.

• Xerostomia: Beyond circadian rhythms, consider the impact of "dry mouth" or overnight mouth breathing on saliva viscosity in the discussion.

• Reference Standards: Discuss whether the lack of correlation stems from SOSM failure, USG limitations (e.g., protein intake, muscle mass), or both methods failing to reflect true plasma osmolality.

Conclusion and References

• Speculation: The statement that SOSM "may hold value under different conditions" is speculative and not supported by the current data; it should be softened or removed.

• Formatting: Ensure the reference list strictly follows PLOS ONE guidelines.

Reviewer #5: Abstract

* The statistical analysis should be included briefly in the methods section on the abstract.

*The values of ICC = 0.00 and κ = -0.02 are quite striking. This suggests that there is no correlation between the SOSM value and USG, which is considered the gold standard.

*The fact that the AUC values range between 0.40 and 0.49 indicates moderate predictive performance; however, the results of the test-retest analysis suggest that this is actually at an excellent level, which is somewhat confusing.

*The conclusions section focuses on the reliability of the device, but the absence of any information regarding its validity is a cause for concern.

Introduction

* The limitations of methods such as plasma osmolality and urine indices in practical applications have been briefly mentioned. However, more detailed information should be provided regarding the underlying causes and processes behind these limitations.

*'In controlled studies', were these studies randomised controlled trials, or did they examine changes in the effects of dehydration during the acute or chronic phases? In particular, if the studies concern conditions below the specified dehydration levels or dehydration induced by exercise, information regarding the use of SOSM should be provided.

*The authors have noted that the MX3 system has previously yielded successful results in elderly and pediatric populations (11, 12), but they could provide a more specific physiological explanation (e.g., exercise-induced changes in saliva composition) as to why these findings cannot be directly applied to healthy and physically active individuals.

*The study specifically chose to measure hydration levels in the morning, and it is advisable to elucidate the physiological reasons behind this decision.

Method

* Prior to morning measurements, it has been stated that participants were not required to follow a specific diet or fluid intake regimen the day before. However, the impact that diet and meal patterns may have on morning hydration levels in elite athletes should not be overlooked. In such cases, if a dietary questionnaire or fluid intake protocol was used, these should be included; otherwise, this limitation should be noted.

*Mouth breathing is common among elite athletes, particularly during sleep. This can reduce saliva flow rate and artificially increase osmolality. The method does not specify whether any observations or screening for this were carried out.

*A 15-minute interval has been allowed between the collection of the urine sample and the saliva sample. Whilst this may seem a reasonable period, it should be noted whether a ‘definitive’ check has been carried out to ensure that no fluids have been consumed during this time.

*The ‘Power Calculation’ section states that 128 matched samples are required to achieve the ICC value. However, samples were collected from only 118 participants in the study. This appears to fall short of the target. Please specify if any data or participants were excluded. If not, please explain how this shortfall in the number of matched samples was addressed.

*The fact that 230 samples were taken from 118 athletes implies that samples were taken from some athletes on more than one occasion (over 2 or 3 days). If an appropriate adjustment (e.g., a mixed-effects model) has been made in the analysis for these ‘within-subject’ correlations (nested data), this should be stated. Otherwise, the p-values and confidence intervals may have turned out to be narrow by chance.

*The measurement of the device’s internal consistency makes sense, but my advice to authors is that they should state that this ensures measurement consistency.

*The SOSM values were categorized using the threshold values provided by the manufacturer (MX3). However, for USG, Casa et al. (15) were used as a reference. It should be noted to what extent these two different sources overlap in order to establish the biological basis for the definitions of 'thirst.' This may have contributed to the finding of ‘negligible agreement.'

Results

*In the study, 36% of participants provided data for three days, while 42% provided data for just one day. This situation may undermine the assumption of ‘independent observations’ in statistical analysis. The authors need to clarify in the statistics section whether they used a ‘mixed-effects model’ for these repeated measurements (nested data).

*The fact that the reported AUC values fall within the range of 0.40–0.49 indicates that, statistically speaking, this test is even less accurate than the threshold of 0.50. It is unclear whether this accuracy is merely random or represents a measurement specific to the device.

*- A kappa value of 0.02 indicates low agreement between the SOSM and USG categories. However, as shown in Table 4, the fact that athletes classified as ‘significantly dehydrated’ based on urine analysis were categorized as ‘hydrated’ or ‘mildly dehydrated’ by the SOSM suggests that the SOSM measurement device yields different results compared to USG. This point requires clarification.

Discussions

*The exclusion of 13 data sets due to ‘high viscosity’ is a factor that undermines the device’s reliability under real-world (field-based) conditions. If the saliva of severely dehydrated athletes is at its most viscous and the device is unable to measure it, the findings may introduce a bias in favor of dehydration. This issue needs to be addressed.

*They have attributed the elevated SOSM values measured immediately upon waking in the morning to the circadian rhythm. However, ‘dry mouth (xerostomia)’ or the effect of breathing through the mouth throughout the night on saliva viscosity could also be added to the discussion.

*The author has used SG (urine specific gravity) as a reference but has also discussed the limitations of USG (muscle mass, protein intake, etc.). If USG is not perfect either and does not correlate with SOSM, the confusion surrounding this situation—namely, whether both methods might be failing to accurately measure true plasma osmolality—should be clarified.

*They stated that they excluded 13 samples from the analysis due to their ‘high viscosity.' The discussion section should address whether this situation creates a ‘selection bias.'

References

* In the Conclusions section, the confusion in this section—where the statement that SOSM ‘may hold value under different conditions’ remains speculative as it is not supported by data—should be clarified.

* The reference list must be formatted in accordance with the PlosOne guidelines.

6. PLOS authors have the option to publish the peer review history of their article (what does this mean?). If published, this will include your full peer review and any attached files.

Reviewer #1: **Yes:** Nathan Bartman

Reviewer #2: **Yes:** Jacobus p Van Wouwe

Reviewer #3: No

Reviewer #4: No

Reviewer #5: No

---

## [Author Response · Author response to Decision Letter 1]

27 Apr 2026

Reviewer 1 (Nathan Bartman)

Thank you for your positive assessment and constructive feedback. We have addressed all of your comments and provide point-by-point responses below.

Comment 1. An important consideration is that this study assessed two different measurement tools for two different body fluids, simultaneously testing device accuracy and the usefulness of saliva to reflect urinary hydration status. Based on these results, it might be more appropriate to label saliva as inconsistent for showing hydration status. In other words, if the device were used to test urine, would it have closer agreement with USG?

Response. We agree that this is an important conceptual distinction and have incorporated it more explicitly into the Discussion. The study simultaneously tests two things: (1) whether the MX3 device reads SOSM consistently (which it does, as shown by the high within-sample ICC), and (2) whether SOSM as a biomarker reflects the same information as USG (which it does not, as shown by the near-zero validity ICC and negligible kappa). We now state more clearly that the poor agreement likely reflects the unsuitability of first-morning saliva as a hydration matrix in this context, rather than device malfunction per se. Regarding the hypothetical of using the device to measure urine: this falls outside the device's intended measurement principle (electrochemical impedance spectroscopy of saliva) and we therefore do not speculate further on this in the manuscript.

Comment 2. Abstract (L30–31): A kappa value <0 might be better described as "no agreement" rather than "negligible".

Response. We agree. A negative kappa value falls below the lowest interpretable category in the McHugh scale. We have revised the abstract to read "no meaningful classification agreement (κ = −0.02; 95% CI [−0.15, 0.10])", which more accurately reflects both the magnitude and the statistical uncertainty of this finding. Because the confidence interval spans zero, we cannot conclude that the value is meaningfully different from chance-level agreement.

Comment 3. Abstract (L74): Did you confirm that participants were fasted, or did you control the amount of time fasted?

Response. Participants were instructed to remain fasted from bedtime until saliva collection was completed, but fasting duration was not formally quantified and adherence was self-reported rather than independently verified. We have added this clarification to the Methods section, and it is also acknowledged as a study limitation.

Comment 4. Introduction (L150, L160): Period appears before the reference.

Response. Corrected in the revised manuscript.

Comment 5. Methods — Hydration Classification: It is unclear whether participants need to fit both SOSM and USG criteria for classification. The USG categories should be defined explicitly.

Response. We have revised the Hydration Classification section to clarify that SOSM and USG categories are assigned independently using separate source-specific thresholds. A participant is classified according to each method separately, not by a joint criterion. The USG categories (well hydrated ≤1.010, minimal dehydration 1.011–1.020, significant dehydration 1.021–1.030, and serious dehydration ≥1.031) are now defined explicitly in the Methods.

Comment 6. Results — Table 2: Some sub-group proportions add up to 100.8%. It is also unclear why USG has both ordinal categories and cumulative binary thresholds, and whether these are mutually exclusive.

Response. We have corrected the rounding in Table 2. The ordinal categories and the binary threshold categories serve different analytical purposes. We have added a table note clarifying that the binary threshold values represent cumulative proportions and are not mutually exclusive.

Comment 7. Discussion: The SOSM threshold for USG ≥1.025 is 76 mOsm, which is lower than the 84 mOsm for USG ≥1.020. Can you clarify why?

Response. The SOSM thresholds were derived empirically from ROC curves by maximizing Youden's index and are therefore data-driven rather than physiologically ordered. The non-monotonic pattern (76 mOsm for USG ≥1.025 vs. 84 mOsm for USG ≥1.020) likely reflects the low prevalence of samples exceeding USG ≥1.025 (13%), which reduces the stability of the Youden-optimized estimate. Given that this pattern does not affect the overall interpretation, as AUC values near 0.50 and kappa values near zero across all thresholds indicate uniformly poor discrimination, we did not add explanatory text to the manuscript.

Comment 8. Discussion (L258): To enable equal comparison with Atjo et al. (r = 0.78), what was the Pearson r value in the present study?

Response. The Pearson correlation between SOSM and USG in our sample was r = −0.01 (95% CI [−0.14, 0.12]), consistent with ICC = 0.00. We have added this value to the existing validity statement in the Discussion to enable direct comparison with Atjo et al.

Comment 9. Discussion (L265): Rephrase "USG cut-offs" as "USG categories" to avoid implying a single threshold.

Response. Corrected. We have revised the relevant passage in the Discussion to read "USG threshold" rather than "USG cut-off" to avoid implying that classification is based on a single value rather than a defined range.

Comment 10. Since MDC80 and MDC90 values were reported, can you explain what these mean to the average user?

Response. We have added a brief clarification to the Results, noting that repeated readings must differ by at least the MDC value to exceed measurement error. Specifically, the text now reads that the MDC90 of 18.4 mOsm indicates that a difference between two repeated readings must exceed this value to be considered a true change rather than measurement error, with 90% confidence.

Comment 11. Discussion (L305): Change "was" to "were".

Response. Corrected.

Comment 12. Limitations: Why was serum (plasma osmolality) not analyzed?

Response. Plasma osmolality was not feasible given that data collection took place at remote international training camps where venous blood sampling and laboratory analysis were logistically not possible. The manuscript notes that plasma osmolality was not feasible in the camp-based setting, and the limitations section now explicitly acknowledges that neither plasma osmolality nor body mass change was assessed, precluding definitive conclusions about whether the discordance reflects failure of SOSM, limitations of USG, or both.

Sincerely,

The authors

Reviewer 2 (Jacobus P. Van Wouwe)

Thank you for your positive appraisal and constructive comments. We address each point below.

Comment 1. It is unclear whether there were any missing values or participants willing to join who actually did drop out.

Response. All 118 athletes who provided written informed consent completed the study protocol; there were no dropouts. Across the 230 sampling occasions, 13 saliva sample sets were excluded because the MX3 device did not return a readable value, and these are now reported explicitly in the Results. No participants withdrew and no urine samples were missing.

Comment 2. The differences in weather conditions across collection sites (Fuerteventura in February, Font-Romeu in October, Stockholm in November) have not been taken into account. Could environmental conditions influence SOSM differently than USG? Are results for samples from the three locations identical?

Response. Since USG and SOSM were collected simultaneously under the same morning conditions at each site, differential environmental effects on the two measures are unlikely to explain the observed discordance — both biomarkers were exposed to the same ambient conditions. Site-specific subgroup analyses were not performed as sport type was confounded with location and the study was not powered for such comparisons. We therefore did not add this as a formal limitation in the manuscript.

Comment 3. In the Results, you report no sex difference in USG. This seems also to be the case for SOSM — why not add it?

Response. We agree and have added the sex comparison for SOSM to the Results. The revised manuscript now states that no significant sex difference was observed in either USG or SOSM (p > 0.05 for both).

Sincerely,

The authors

Reviewer 3

Thank you for the thorough and thoughtful review. Your twelve questions have helped us sharpen several aspects of the manuscript. We respond to each question below.

Question 1. Why was urine specific gravity selected as the sole reference method, given that it is not considered the gold standard?

Response. We acknowledge that USG is not the gold standard. However, plasma osmolality was not feasible given the absence of laboratory infrastructure at the remote training venues. USG was selected because it is the most widely used field-based hydration reference in applied sport settings, recommended by major sports medicine organizations for first-morning monitoring, and supported by prior validation against plasma osmolality. We have added an explicit justification for this choice in the Discussion, along with an acknowledgment of USG's own limitations in athletic populations.

Question 2. Did the authors consider including plasma osmolality or body mass changes as additional validation markers?

Response. Yes, we considered both. Plasma osmolality was not logistically feasible, as described above. Body mass change was not measured because the study focused on first-morning classification of hydration status rather than acute fluid balance tracking, and because reference body mass in the euhydrated state was not available for these athletes at the time of measurement. Both omissions are acknowledged in the limitations section of the revised manuscript.

Question 3. How was the potential influence of circadian rhythm on salivary osmolality controlled?

Response. Circadian variation was standardized by design: all samples were collected within a fixed morning window (06:00–08:00) under fasted conditions, minimizing within-individual temporal variation. However, we cannot exclude that circadian effects — such as reduced salivary flow and elevated osmolality upon waking — inflated SOSM values systematically in the morning. We discuss this explicitly in the Discussion as a potential explanation for the high prevalence of SOSM readings classified as dehydrated relative to USG.

Question 4. Could the lack of intentional hydration manipulation have limited the range of observed values and thereby affected diagnostic validity?

Response. Yes, this is discussed in the Discussion. Free-living conditions maximized ecological validity but resulted in a cohort that was generally well-hydrated: only 13% exceeded USG ≥1.025. A restricted range of hydration states reduces the statistical signal available for discrimination, which likely contributed to the poor diagnostic accuracy of SOSM in this sample. This is contrasted with the Atjo et al. cohort, where 89% exceeded USG 1.025 and where stronger agreement was reported.

Question 5. How do you interpret the finding that most samples were classified as dehydrated by SOSM while USG indicated normal or mildly reduced hydration?

Response. We interpret this as reflecting circadian and local oral factors rather than true whole-body dehydration. Salivary flow is at its diurnal nadir in the morning and solute concentration is elevated, which may inflate SOSM independently of systemic hydration. Overnight oral drying from mouth breathing, snoring, or xerostomia may further concentrate saliva locally. These factors are discussed in the physiological explanation section of the Discussion.

Question 6. Is it possible that discrepancies arise from physiological differences in saliva and urine regulation rather than purely methodological limitations?

Response. Yes, and we now state this explicitly. USG reflects renal concentration of solutes over the preceding hours, whereas SOSM reflects the composition of oral fluid at the moment of sampling, regulated by salivary gland secretion, circadian rhythms, sympathetic tone, and local oral conditions. These are physiologically distinct compartments, and we clarify in the Hydration Classification section and Discussion that agreement was assessed as an empirical question rather than assumed on the basis of biological equivalence.

Question 7. Did the authors conduct subgroup analyses by sex, sport discipline, or performance level?

Response. Formal subgroup analyses by sport discipline and performance tier were not conducted. There is no established physiological mechanism by which SOSM validity would be expected to differ systematically across sport types or competitive levels, and the number of athletes in several sport-specific subgroups was too small for meaningful analysis. Sex differences in both USG and SOSM were examined and are now reported in the Results; no significant differences were found for either measure.

Question 8. To what extent might saliva viscosity and unsuccessful device readings have influenced the results, and was a sensitivity analysis performed?

Response. The 13 excluded sample sets represent cases where the MX3 device returned no readable value, most plausibly due to elevated saliva viscosity. We discuss the potential for selection bias explicitly: if highly viscous or concentrated saliva was disproportionately unreadable, the exclusions may have removed precisely those cases where the device would perform most poorly. A formal sensitivity analysis was not possible because the excluded cases lack SOSM values. The potential direction of this bias is acknowledged in the Discussion and limitations.

Question 9. Are the manufacturer's SOSM classification thresholds appropriate for elite athletes, or should they be calibrated specifically for this population?

Response. It is physiologically unlikely that fasted morning saliva composition in elite athletes would differ substantially from that of other healthy young adults, suggesting that the manufacturer thresholds themselves may not be the primary explanation for the poor classification agreement. Rather, our findings suggest that first-morning saliva is an unsuitable matrix for hydration classification in this context, regardless of the thresholds applied. Population-specific validation under different sampling conditions would be needed before any practical application in athletes could be justified.

Question 10. How do the authors envision potential applications of SOSM under alternative conditions, such as post-exercise or during training?

Response. We deliberately avoid speculating about alternative conditions given the absence of supporting data. The Conclusion states that future studies should examine SOSM under other sampling conditions before any practical role can be inferred.

Question 11. Could repeated measurements from the same saliva sample have artificially inflated the reliability estimates compared to real-world use?

Response. Yes, and we now make this distinction explicit in both the Results and Discussion. Repeated readings of the same sample eliminate variation from collection technique, volume, and transient oral factors. Our ICC of 0.93 therefore represents the upper bound of device repeatability under optimal conditions, as we acknowledge by contrasting it with the lower ICC (0.75) reported by Winter et al., who collected two separate saliva samples.

Question 12. What are the practical implications for coaches and medical staff who already use the device?

Response. The principal practical implication is that first-morning SOSM measured with the MX3 HTS should not be used as a standalone hydration indicator in elite athletes under free-living conditions. The device can read SOSM consistently, but the values do not correspond to USG-based hydration categories in this population and context. We have reinforced this implication in the Conclusion.

Sincerely,

The authors

Reviewer 4 (reviwer report.pdf)

Thank you for your careful reading and the detailed and structured evaluation. We have revised the manuscript substantially in response to your suggestions. Below we provide a point-by-point response describing how each comment has been addressed.

Comment 1. Please briefly include the speci

---

## [Decision Letter · Decision Letter 1]

8 May 2026

Salivary Osmolality Measured by MX3 Hydration Testing System Demonstrates High Reliability but Limited Validity in Elite Athlete Hydration Assessment

PONE-D-25-59162R1

Dear Dr. Pettersson,

We’re pleased to inform you that your manuscript has been judged scientifically suitable for publication and will be formally accepted for publication once it meets all outstanding technical requirements.

Kind regards,

Ozkan Isik

Academic Editor

PLOS One

Additional Editor Comments (optional): Thank you for your interest in the article revisions.

Reviewers' comments:

Reviewer's Responses to Questions

**Comments to the Author**

1. If the authors have adequately addressed your comments raised in a previous round of review and you feel that this manuscript is now acceptable for publication, you may indicate that here to bypass the “Comments to the Author” section, enter your conflict of interest statement in the “Confidential to Editor” section, and submit your "Accept" recommendation.

Reviewer #2: All comments have been addressed

Reviewer #3: All comments have been addressed

2. Is the manuscript technically sound, and do the data support the conclusions?

Reviewer #2: Yes

Reviewer #3: Yes

3. Has the statistical analysis been performed appropriately and rigorously? 

Reviewer #2: Yes

Reviewer #3: Yes

4. Have the authors made all data underlying the findings in their manuscript fully available?

Reviewer #2: Yes

Reviewer #3: Yes

5. Is the manuscript presented in an intelligible fashion and written in standard English?

Reviewer #2: Yes

Reviewer #3: Yes

6. Review Comments to the Author

Reviewer #2: This revised manuscript is in my opinion now suitable for publication as most comments have been adequately answered.

Reviewer #3: Thank you for your careful revision of the manuscript. The updated version clearly reflects that the authors have addressed the reviewer’s comments in a constructive and comprehensive manner. The methodological clarifications, particularly regarding the choice of reference method and the interpretation of validity results, have improved the transparency and scientific rigor of the study. In addition, the revisions to the discussion section provide a more balanced interpretation of the findings and better acknowledge the limitations of the approach. The manuscript is now more coherent, and the conclusions are more appropriately aligned with the presented data. Overall, these changes have strengthened the quality and clarity of the work, and the revised version represents a meaningful improvement over the original submission.

Kind regards,

Reviewer

7. PLOS authors have the option to publish the peer review history of their article (what does this mean?). If published, this will include your full peer review and any attached files.

Reviewer #2: **Yes:** Jacobus P Van Wouwe

Reviewer #3: No

---

## [Editor Report · Acceptance letter]

PONE-D-25-59162R1

PLOS One

Dear Dr. Pettersson,

I'm pleased to inform you that your manuscript has been deemed suitable for publication in PLOS One. Congratulations! Your manuscript is now being handed over to our production team.

Kind regards,

on behalf of

Professor Ozkan Isik

Academic Editor

PLOS One